# Everyday life in a Swedish nursing home during the COVID-19 pandemic: a qualitative interview study with persons 85 to 100 years

Qarin Lood ,[1,2] Maria Haak,[3,4] Synneve Dahlin-Ivanoff[5]

For numbered affiliations see end of article.

**Correspondence to**
Dr Qarin Lood;
qarin.lood@neuro.gu.se

## ABSTRACT

**Objective** To understand and report on the impact of the COVID-19 pandemic on the everyday lives of frail older persons living in nursing homes by exploring their experiences of how the pandemic-related restrictions had influenced them and in what way.

**Design** Empirical qualitative interview study.

**Setting** A publicly run nursing home in an urban area in Sweden in June 2020. The nursing home had visitor restrictions, cancelled activities and physical distancing requirements since March 2020.

**Participants** A total of 10 persons, 85–100 years, living in a Swedish nursing home during the COVID-19 pandemic, were recruited through nursing home management and interviewed in June 2020 using medically approved visors and physical distancing.

**Analysis** Interviews were analysed using thematic analysis, which involves familiarisation, coding and definition of themes. Transcripts were coded into data-driven categories before being organised into categories that described and explained the data.

**Results** The analysis resulted in the main theme 'It is like living in a bubble', that describes everyday life in the nursing home during the pandemic as a world of its own in which the older persons felt both protected and isolated. This is described in four subthemes: living 1 day at a time, without fear of the virus; feeling taken care of; having limited freedom and missing out on the little extras.

**Conclusions** Contributing to the growing area of COVID-19-related research, our findings provide novel insights into how pandemic-related restrictions in nursing homes represent a risk of isolating older people from the outside world and diminishing their freedom. Put in relation to the previous research, these findings could be applied beyond the pandemic, to develop research and practice that puts focus on how to support older people to decide for themselves how to spend the rest of their lives.

## INTRODUCTION

The novel coronavirus that is causing COVID-19 has changed pretty much everything that people do. It spreads rapidly and could cause severe and fatal infections, especially among people living in nursing homes, who are often experiencing physical frailty[1] and compromised physiological barriers.[2] In

### Strengths and limitations of this study

► The qualitative approach provides novel insights into how the COVID-19 pandemic has influenced people living in nursing homes.

► The first-hand experiences described complement the medical understanding of the impact of COVID-19 on older people.

► A limitation of the study is the risk of biased analysis due to the lived experiences of the pandemic among the researchers themselves.

Sweden, person living in nursing homes are typically living with multiple health problems and are in need of access to care staff round-the-clock.[3] As per 4 April 2021, a total of 16 204 persons living in Swedish nursing homes had been confirmed with COVID-19 and 5446 (34%) of them had died with the disease. This constitutes 43% of the amount of people who had died with COVID-19 in Sweden at that time.[4] People in nursing homes are one of the most affected groups globally, which can be seen from the research produced.[5–8] Out of fear of the virus, nursing home organisations around the world have made drastic changes to their services to diminish the spread of infection in line with the World Health Organization's (WHO) international guidelines.[9] In line with these guidelines, Swedish nursing homes implemented physical distancing and visitor restrictions, and most organised activities were cancelled from 30 March until 1 October 2020. Although not being a ban, which is not supported by Swedish law, the visitor restrictions initiated a public debate on the risk of social isolation and the negative consequences that might have,[10] and they were lifted in October 2020 based on the risk of negative consequences of isolation instigated by lengthy visitor restrictions.[11] So far, research has mainly focused on family members' experiences of the

restrictions,[12 13] and the impact they may have on people living in nursing homes is poorly understood, with no primary data on their experiences.

Already before the pandemic, there were reports on social isolation, limited quality of life and near endemic loneliness among people living in nursing homes.[14 15] This poses serious threats to their everyday fulfilment and sense of dignity.[16–18] The pandemic tends to increase those threats,[19] since restrictive measures taken to protect people residing in nursing homes from infection may have a negative impact on their well-being.[20] For instance, social isolation is a serious public health concern that may lead to medical illness such as cardiovascular diseases,[21] and mental health problems, such as loneliness, depression and cognitive decline.[22] There are also recent reports on the negative impact of social disconnection and perceived isolation on both anxiety and depression among older persons.[23]

According to the Swedish government,[24] a modern society and welfare state needs to improve the possibilities for all older people to participate in society, to enjoy a decent quality of life and to experience health, meaningful activities and social participation. Research on best practice nursing home care also puts focus on activities to promote thriving and support a good quality of life,[25–27] and there is evidence to suggest that quality of life is improved by participating in personally meaningful activities.[28] Yet, as part of one of the risk groups for severe disease or death from COVID-19, persons living in nursing homes have faced restrictions that have resulted in no visitors, communal dining or leisure activities. The scientific community is responding rapidly to the consequences of those restrictions, but what has been left in the dark is how the persons in nursing homes themselves experience the measures taken to protect them. Therefore, this paper aimed to understand and report on the impact of the COVID-19 pandemic on the everyday lives of frail older persons living in nursing homes by exploring their experiences of how the pandemic-related restrictions had influenced them and in what way.

## METHODS

Seeking to understand the complexity of everyday life in a residential care home during the COVID-19 pandemic, this study had a qualitative design to explore and describe older persons' experiences. Data were collected from 10 persons in a nursing home over a 2-week period in June 2020. The authors of the study are all registered occupational therapists, with research expertise in gerontology, occupational science and health science.

### Patient and public involvement

As part of a larger research programme on user involvement in research,[29] the research questions for this study were discussed in seminars with user representatives from pensioner associations in Sweden before commencement. Drawing on the findings from this study, the user representatives will also be involved in future planning of research on user involvement in research on ageing and health, together with researchers and healthcare representatives from different disciplines and professions. The findings of the study will also be presented to interested participants in dialogue with the first author or through a short popular scientific report. They will also be presented with the choice to read the scientific report.

### Study setting

Participants were recruited from a nursing home with 101 beds (27 in dementia care units) and about 85 staff members. Four of the people living in the nursing home had contracted the virus, all of them survived and were well at the time of data collection (two of them were included in the study). A few staff members had also contracted the virus and had been, or were, on sick leave. Care staff were available for the people living there 24 hours, 7 days a week. Registered nurses, physicians and allied health professionals were generally available for care and rehabilitation when needed, but during the COVID-19 pandemic, such visits were confined to acute needs. Only outdoor visits from friends and family were allowed during the study period, with a maximum of two visitors at a time. Dates and times for visits were organised by nursing home managers, and staff escorted the older persons back and forth to the outdoor area. Indoor visits were allowed for special reasons only, such as end-of-life-care or severe anxiety. Delivery of food or things from outside the nursing home had to be delivered through staff. All staff were entitled to adhere to the recommendations by the Public Health Agency of Sweden; they were strongly encouraged not go to work when feeling ill, even with mild symptoms, and to follow basal hygiene routines such as hand hygiene and disinfection of areas in contact with human beings.

### Participants and data collection

All persons assessed by the nursing home staff as cognitively able to give informed consent and to hold a conversation for at least 15 min were invited to participate through receiving general written information about the study from nursing home managers. Eleven persons expressed interest and time and place were set for the interview. All potential participants received detailed written and verbal information about the study from researchers before the interviews and had the opportunity to ask questions about the study and their participation. Every effort was made to ensure that the choice to participate, and to what extent, was the person's own, not limited by research approaches or structures. One person withdrew their consent after the interview had been conducted.

The participants were between 85–100 years of age (85, 86, 89, 90, 94, 96, 98, 100, 100 and 100 years, respectively). All were physically frail[30] and in need of support in activities of daily living. None of the participants had been diagnosed with dementia. Seven participants were women, eight had children and three of them had partners who

were still alive. All data collected from the person who withdrew their consent were deleted and not included in the study, and no other person than the researcher who conducted the interview took part of the interview. The interviews focused on how the older persons' everyday life had been influenced by the pandemic and their increasingly frail bodies, and they lasted between 17 and 60 min. Medically approved visors were used in all contacts between researchers and participants and a distance of at least two metres was kept at all times. All interviews were recorded digitally and transcribed by the first author before analysis.

### Data analysis

Triangulation[31] was applied through constant comparative analysis to address trustworthiness. This meant that data were analysed by all three authors using the stepwise procedure for thematic analysis[32] as follows. First, all authors listened to the interviews repeatedly to become familiarised with the data. Second, initial codes that described the participants' experiences in a condensed way were generated by the second author, who had not been involved in the data collection, and discussed among all authors. Third, the initial codes were revised by the first author to search for themes in a process of listening to and reading all collected data again. Notes were taken on essential meanings and extracted data were interpreted and organised into a prospective thematic structure by the first and third authors. Fourth, the prospective themes were reviewed and all authors discussed relationships between codes, themes and different levels of themes to define them and assess the validity of each prospective theme in relation to the data set as a whole. Considering internal homogeneity and external heterogeneity,[33] phase 4 also involved a refinement of themes, and the fit of data extracts within each theme was assessed. This process was conducted in close collaboration between the authors. Discrepancies were discussed and a final decision was made by the first author who was responsible for the analysis. Fifth, all data extracts were interpreted to define final themes based on the meaning and implications of the data. This process continued until it was not considered possible to conduct any further refinements of the themes. Sixth, the scientific report was written by all authors.

## RESULTS

The participants' experiences were interpreted into one main theme and four subthemes. The interpreted experiences are supported by quotations from a selection of participants.

### It is like living in a bubble

The overarching interpretation of the participants' experiences of everyday life in the nursing home during the COVID-19 pandemic was that it was somewhat a world of its own. The pandemic-related restrictions made the participants feel safe and secure in terms of virus transmission and support in activities of daily living, but at the same time isolated from the outside world. This is described in four subthemes: *living 1 day at a time without fear of the virus, feeling taken care of, having limited freedom* and *missing out on the little extras.*

### Living 1 day at a time, without fear of the virus

Participants handled the pandemic and their ageing bodies by living 1 day at a time, and when being asked whether they were afraid of the virus, they expressed that there was no need to worry of getting infected since they were so old. The opportunity to live 1 day at a time was facilitated by the nursing home bubble in terms of protection from virus transmission and the support provided by nursing home staff made it possible to seize the day and enjoy the activities that could proceed despite the pandemic. For instance, taking walks in the corridors, read books, do cross-words and strive to carry out personal care activities independently to maintain one's physical and cognitive abilities. Thus, even if the ageing and physically frail body represented an insecurity as to whether pandemic-related changes to daily routines would be possible to handle, the participants experienced that they could choose to focus on, and appreciate, the small things in life and live 1 day at a time, rather than worrying for what might happen in the future. This was also described in terms of having no choice but to accept the situation, even if being old and frail was experienced as tough. A 94 year old woman who had survived COVID-19 described this as:

> I am very tired now, because I have had this, Corona. I believe it was not too bad for me. I had a fever and they stated that it was that (Corona), but I wasn't. I had a strange feeling in my body and felt miserable, but I don't think I had the worst. Breathing is difficult anyway, cancer operation. Right now, it seems to be alright. I have other cancer as well. It has been a lot. But it's alright, you have to live one day at a time.

### Feeling taken care of

Living in a nursing home during the COVID-19 pandemic contributed to a feeling that one was taken care of, both in terms of being protected from the virus and being cared for by staff. Daily routines could still be carried out, representing an important part of everyday life that was not affected by the pandemic, and the nursing home staff embodied a much-appreciated human contact in times of social isolation. Staff provided support for the participants to defy both the pandemic in terms of protection from virus transmission, and the ageing body in terms of support to maintain essential activities and daily routines. This contributed to an everyday life that, in many ways, was the same as before the pandemic. Staff also offered a sense of security in knowing that there would be someone there to care for you and provide support when needed. This is visualised by a quotation by an 89 year old man's

response to the question about how they felt about living in the nursing home during the pandemic:

> I am very satisfied. Because of the staff, that they are so helpful. They are helpful and come in and talk with me. I have nothing to complain about… I cannot manage on my own, here I am spoiled, here I get food and the assistance that I need.

### Having limited freedom

Everyday life in the shielded world of a nursing home during the pandemic involved not having the freedom to choose for oneself what to do, with whom and when. Instructions from nursing home staff and authorities were dutifully followed, even though the participants did not always agree with them. There was a perception that there simply was no option, and there were expectations on authority representatives to set a date for when the isolation would be over and everything would get back to as it was before the pandemic. The limited freedom also involved increased dependency on staff, and the participants became more inactive than before the pandemic. They experienced that their health deteriorated, both because of age and frailty, and because of the pandemic-related restrictions with few opportunities to move about, to go outside or to receive visitors without assistance from staff. Envisaging life after the pandemic, participants dreamt about moving around freely again and regaining their strength and energy after being passive due to the restrictions or after having COVID-19 themselves. The feeling of having limited freedom in everyday life is illustrated by a 100 year old woman's answer to the question: What is the biggest difference for you since the pandemic started?

> The freedom. I am dependent you see. So, I cannot choose… Nobody is allowed to go out, nobody is allowed to come in.

### *Missing out on the little extras*

Everyday life during the pandemic was experienced as empty and dull, without the things that were considered the little extras that provided a silver lining to everyday life. With no opportunities to go out and do errands, participants were dependent on relatives to bring them the little extra things that they could not get in any other way, for example, fresh flowers, bakery or new clothes. Initiatives had been taken to deliver things from relatives to the participants through staff, but it was not always possible due to staff being under extra pressure during the pandemic. For this reason, participants did not want to disturb staff with things that might seem trivial, such as having a chat or to go out for a walk, and because no visitors were allowed, life was experienced as being somewhat destroyed by the pandemic-related restrictions. Everyday life in the nursing home came to circle around daily routines and news on the pandemic, and the telephone came to be one of few contacts with the outside world.

Although not being able to replace physical contact with friends and family, the telephone gave opportunities to think and talk about other things than the pandemic. The participants also missed doing fun and meaningful activities such as excursions or organised social and creative activities, and even if they understood why these types of activities were cancelled to minimise virus transmission, they lacked the stimulation they received from them. A 96 year old woman described the difference between everyday life before and during the pandemic as:

> (Before Corona) there were (visitors) every day almost. And there were activities in the activity room. We had such a nice time down there, we had coffee at eleven in the joint room, we had a nice time. We sang together and had quizzes, and watched tv together, and we took walks outdoors. And we got to go to the horticultural society (gardens) and now we are not allowed to do anything.

## DISCUSSION
### Statement of principal findings

To the best of our knowledge, this study is the first to explore experiences of everyday life in a nursing home during the COVID-19 pandemic. Illustrated through the metaphor of living in a bubble, everyday life at the nursing home during the COVID-19 pandemic meant living a protected but isolated life, and the findings highlight the need for nursing homes to both aim to protect and sustain physical health and capacity and to support psychosocial aspects of life. The illumination of how frail older people in nursing homes experienced the pandemic-related restrictions present serious challenges to older people's freedom to do what they have a reason to value. Thus, by giving voice to the older persons themselves, the present findings provide insights into how they would have wanted everyday life during the pandemic to be. Although not expressing a willingness to die, the participants were not afraid of contracting the virus, and they wished to decide for themselves how to spend the rest of their lives, rather than being limited by restrictions to minimise virus transmission. As Bayer[34] stated already in 2007, there are continuing tensions between individual rights and public health, and the question he poses on the extent of protection of public welfare in relation to fundamental rights of individual people is perhaps even more relevant now than ever before. As Bayer points out, authoritarian measures to protect the public may be sanctioned by the notion of common good. However, even if the pandemic situation in 2020 required drastic measures, it seems as if authorities were given perhaps too much liberty to diminish virus transmission, with sometimes dire consequences for people approaching the end of life.

### Strengths and weaknesses of the study

The major strength of this study is the uniqueness of the qualitative data gathered from a sample of persons 85 years and older living in nursing homes during the

COVID-19 pandemic, giving voice to a seldom heard, but largely affected group of people. The findings, thus, provide empirical support for what until now have been assumptions by other people than the older persons themselves.[20 35] Consequently, even if the number of participants was limited due to the restrictions applied, the first-hand experiences described provide important insights into the field of nursing home care in the light of the pandemic. Moreover, as described by Malterud,[36] rigour in qualitative studies depends on reflexivity and not a specified sample size[36] and although representing a Swedish example, the similarities between restrictions implemented in nursing homes around the world increase the international value of the study. The findings complement the medical understanding of the impact of COVID-19 on frail older people with an understanding of how the pandemic-related restrictions may influence older people's everyday life and well-being. Due to the qualitative design, however, it is impossible to draw any conclusions on the impact of the restrictions or the pandemic on the physical and mental health of older persons in nursing homes. Another limitation with the study is the fact that the researchers were studying experiences of the pandemic, while at the same time experiencing themselves. Even if the authors strove to be aware of their own expectations and experiences throughout the analysis procedure, this may have resulted in a biased analysis. Every attempt was, however, made to assure that the authors' experiences did not overshadow the voices of the older persons participating in the study.

### Other studies

Interpreted in relation to previous research on autonomy among older people in nursing homes,[37] the present findings visualise a risk for measures taken to protect people in nursing homes from COVID-19 to become paternalistic. Authorities and clinicians alike may believe that they know best, and older persons' interests and experienced needs may be overridden by others. Supported by what has been described by Dichter et al,[35] the findings call for a balance between infection control and person-centred care that combines biomedical knowledge of diseases with personal understanding of individual person's desires and needs. The older persons' descriptions of everyday life in a nursing home during the COVID-19 pandemic illuminate issues in terms of caring for older people as persons, and there is a need, that goes beyond the pandemic, to implement person-centred care that can support older persons in nursing homes to both remain safe and preserve their dignity and personhood by protecting them from physical threats to their health while, at the same time, attending to their psychosocial needs. As stated by Naldemirci et al,[38] person-centred care involves the responsibility of people working in healthcare to meet each person's needs by personalising the care and prioritise shared decision-making.[38] The COVID-19 pandemic has challenged this endeavour and highlighted a need for

infection control interventions to also provide psychosocial and mental health support.[39] Deepening the understanding of the negative consequences of the pandemic on older persons' everyday lives in nursing homes, the present findings, thus, provide a foundation for future research to attend to health and well-being in a broader sense than strict infection control.

The present findings also contribute with an understanding of how pandemic-related restrictions in nursing homes may be experienced as isolation and deprivation of freedom of choice. Throughout the pandemic, visitor restrictions have been implemented without scientific support for the effect of such measures to minimise virus transmission,[40] and there is a lack on primary data on how this may impact the lives of persons living in, and visiting, nursing homes. This study, thus, presents a unique contribution to the growing area of research on the COVID-19 pandemic, illustrating how the pandemic-related restrictions may be as great a threat to the health and well-being of older people in nursing homes, as the virus. As described by Landry et al,[41] public health measures to reduce transmission of COVID-19 among older persons need to attend to several dimensions of health[41] and contradictions between protection from the threats of COVID-19 infection and the risk of social isolation imposed by the pandemic need to be attended to.[20 35 42 43] What the present study adds to this understanding of the impact of the COVID-19 pandemic on older persons in nursing homes is the first-hand perspective of representatives from a group identified as being among the most vulnerable when it comes to COVID-19 infection. Contrary to expectations, the findings illustrate how the older persons maintained many daily routines, without fear of the virus, and that they avoided pandemic worry by living 1 day at a time. This has been described in previous research as a strategy applied by persons approaching the end of life[44] and could be understood as a potential for health. In the light of these findings, it seems reasonable to question whether the interventions to minimise the threat of COVID-19 in nursing homes are adequate, and whether mortality is a sufficient outcome to measure the effects of such interventions.

### Implications of the findings and future research

Giving voice to persons representing one of the most vulnerable groups in terms of COVID-19 infection, this study contributes with a unique perspective on the impact of the pandemic, and not the virus per se, on the older persons' lives. Specifically, our study contributes with a visualisation of how restrictions to minimise the risk of infection may not be a sufficient intervention during the pandemic and beyond. Contributing to the growing area of COVID-19-related research, the findings provide a foundation for future research and practice, calling for services that can balance protection from physical threats with psychosocial support.

**Author affiliations**
¹Department of Health and Rehabilitation, Institute of Neuroscience and Physiology, Sahlgrenska Academy, Centre for Ageing and Health—AgeCap, University of Gothenburg Sahlgrenska Academy, Goteborg, Sweden
²School of Nursing and Midwifery, College of Science, Health and Engineering, La Trobe University, Melbourne, Victoria, Australia
³Research Platform for Collaboration for Health, Faculty of Health Science, Kristianstad University, Kristianstad, Sweden
⁴Department of Health Sciences, Lund University Faculty of Medicine, Lund, Sweden
⁵Department of Psychiatry and Neurochemistry, Institute of Neuroscience and Physiology, Sahlgrenska Academy, Centre for Ageing and Health—AgeCap, University of Gothenburg Sahlgrenska Academy, Goteborg, Sweden

**Acknowledgements** First and foremost, we would like to thank the persons participating in the study, sharing their experiences of their everyday lives during these challenging times. We would also like to thank the management at the nursing home for allowing us to conduct the interviews, and the care staff for assisting us with selecting participants and for their tireless commitment to providing good care and support to the persons living in the nursing home. Finally, we thank The Swedish Research Council for Health, Working Life and Welfare (FORTE) who funded the study.

**Contributors** QL designed and directed the project and initiated contact with the management at the nursing home. She also did the literature search and drafted the manuscript. QL and SD-I collected the data and were responsible for data analysis with input from MH. SD-I also contributed with intellectual input on the manuscript draft. MH was responsible for the first steps of the analysis and contributed with intellectual input to the final interpretations as well as the manuscript draft. All authors have reviewed and agreed to the draft and final version of the paper.

**Funding** This work was supported by The Swedish Research Council for Health, Working Life and Welfare (FORTE), grant number 2018–00904.

**Competing interests** None declared.

**Patient consent for publication** Not required.

**Ethics approval** Ethical approval was obtained from The regional ethical board in Gothenburg (ref. no. 813–18) and all participants provided written informed consent before the interviews.

**Provenance and peer review** Not commissioned; externally peer reviewed.

**Data availability statement** Data are available upon reasonable request. The qualitative data generated and analysed as part of the current study are not publicly available due to the information provided to the participants when obtaining their informed consent, stating that all attempts would be made to maintain confidentiality. De-identified data are, however, available upon reasonable request to enable review, and will be stored for 10 years at the University of Gothenburg. All data are covered by the Public Access to Information and Secrecy Act (offentlighets- och sekretesslagen) and a confidentiality assessment (sekretessprövning) will be performed at each individual request. Permission from University of Gothenburg, the Institute of Neuroscience and Physiology, has to be obtained before data can be accessed.

**ORCID iD**
Qarin Lood http://orcid.org/0000-0002-6187-0929

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
