## [Reviewer comments · BMJ Open]

ARTICLE DETAILS

TITLE (PROVISIONAL)	Everyday life in a Swedish nursing home during the COVID-19 pandemic: a qualitative interview study with the oldest old
AUTHORS	Lood, Qarin; Haak, Maria; Dahlin-Ivanoff, Synneve

VERSION 1 – REVIEW

REVIEWER	Rapaport, Penny University College London, Division of Psychiatry
REVIEW RETURNED	05-Feb-2021

GENERAL COMMENTS	Thank you for the opportunity to review this important article, it is impressive that you have been able to conduct and write up this research over a short time frame and as you highlight, it may be one of the first qualitative studies to explore the lived experience of older people in care homes during the pandemic. As such, this has relevance and resonance beyond the local context however I feel that there are a number of revisions that would improve the quality and potential contribution of this paper to the literature. Introduction: p.4 line 12 add in a reference to justify the comment that persons in nursing homes are most widely affected globally. More generally in the introduction it would be good to refer to more of the research literature that has been published about the impact of Covid 19 on residents in nursing homes, this will provide more context to the research being presented. Methods: P.5 Line 14 (PPI section) The sentence starting “Participants will also have the opportunity to...” is unclear – I was not sure what this was referring to so would be good to explain in a bit more detail. p.5 line 35 – It is interesting that two of the participants had survived Covid 19 however you do not refer to this again in the paper. How did their experiences differ if at all, did they refer to having had the virus – perhaps you could explore this further in results and discussion. p.5 line 49 (Participants and data collection) – Please could you give more detail on how the sample was obtained – It seems that this was a convenience rather than a random sample and it is not clear whether all eligible participants were approached and how many of them said yes. Did the participants all have capacity to give informed consent – How did you ascertain whether residents had capacity please could you give more detail here. Some of the results presented (such as the ages and description of the background and demographics of the people interviewed
---

	would perhaps work better as part of the results (separated from the methods as you are describing the participants rather than what you did) and it would work well at the start of your results to situate your sample. Also, as you are interested in frailty and the ageing body, could you describe the sample a little more, did they have dementia or other long term conditions, again this provides useful context for your results. p.6 Data analysis section – How did you resolve discrepancies between the different researchers conducting the thematic analysis if there were any? Results: Overall in reading the results I was disappointed that there was a lack of detail and depth in relation to the data you present. For each of the subthemes you make interesting assertions however these do not seem to be backed up by quotations from the participants and where there are quotations they do not necessarily seem to elucidate the richness that you allude to in your narrative account. This is the case for all of the subthemes and was my main frustration when reading the manuscript. I was struck that you make comments like “this made it possible to defy both the pandemic and the ageing body” but it is not supported by evidence. You refer to frailty and the body at various points but I think need to give a richer, more in depth analysis or if the data is not available to support your assertions then it should be taken out. I was interested in how the subthemes “feeling taken care of” and “being in the hands of others” related to each other – Again I wanted to hear more about these different positions, presumably the participants were referring to staff here taking this dual role of both supporter and enforcer. Perhaps you could say something more about this. Discussion: Your discussion is interesting and you give a good summary of the strengths and weaknesses. p. 8 line 30 “Another question mark is whether the pandemic presents an actual risk to the health of older persons in nursing homes” I may have misunderstood but I am not sure what is meant by this comment – perhaps it could be reworded or explained differently as I think in the introduction you present evidence of the disproportionate impact of the pandemic upon nursing home residents both in Sweden and Globally. I think some of your assertions and interpretation is again limited by the lack of richness in your thematic analysis. For example how do your findings relate to the assertion you make about Dichter et al study – these opposite ends of the spectrum seem to relate to two of your quotes but how do they relate to each other. p.9 line 30 “Contrary to expectations, the findings illustrate how the older persons maintained many daily routines, without fear for the virus, and that they avoided pandemic worry by living one day at a time.” – This may well be because there were tight restrictions in place and that they were able to live without fear because of the restrictions in place. I agree with your comment however I think that perhaps you need to relate this to the particular context of your study.
--	--

REVIEWER	Novo-Veleiro, Ignacio
-----------------	-----------------------

	Complejo Hospitalario Universitario de Santiago de Compostela, Internal Medicine
REVIEW RETURNED	20-Feb-2021

GENERAL COMMENTS	The authors present a manuscript of some interest but it is not really a research paper but really a summary of 10 interviews to people living in a nursing home. It only includes 10 patients, which can not represent the main opinion of people living in the selected nursing home. Even as an interview report, this number of participants is too low to consider it as a valuable paper. The authors should include more patients, at least a half of the whole people living in the center to establish any valid conclusion. The manuscript has a structure of an original research item but the results section is only a summary of the opinions of the interviewed people, it does not include any analysis or data that can lead the reader to any conclusion about their research. In my opinion the manuscript can not be published in the present form and could be rewritten as a letter to the editor or brief communication, since it does not fit as a research paper. The conclusions are unclear and anyway very difficult to sustain with data from only 10 people.
--

VERSION 1 – AUTHOR RESPONSE

Reviewer: 1

Dr. Penny Rapaport, University College London

Comments to the Author:

Thank you for the opportunity to review this important article, it is impressive that you have been able to conduct and write up this research over a short time frame and as you highlight, it may be one of the first qualitative studies to explore the lived experience of older people in care homes during the pandemic. As such, this has relevance and resonance beyond the local context however I feel that there are a number of revisions that would improve the quality and potential contribution of this paper to the literature.

Introduction:

p.4 line 12 add in a reference to justify the comment that persons in nursing homes are most widely affected globally.

Thank you for this suggestion. The sentence has been revised and with the ambition to illuminate the global concern in this matter, literature from the Nordic countries (Sweden), US, and Asia has been added on page 1, paragraph 1.

More generally in the introduction it would be good to refer to more of the research literature that has been published about the impact of Covid 19 on residents in nursing homes, this will provide more context to the research being presented.

Up to date research literature about the impact of COVID-19 on people living in nursing homes has been added. All in all, seven newly published papers with a specific focus on COVID-19 have been added on page 1.

Methods:

P.5 Line 14 (PPI section) The sentence starting “Participants will also have the opportunity to...” is unclear – I was not sure what this was referring to so would be good to explain in a bit more detail. We have revised the sentence to clarify how the participants will be able to take part of the results, should they wish to do so. The revision can be found on page four, second paragraph.

p.5 line 35 – It is interesting that two of the participants had survived Covid 19 however you do not refer to this again in the paper. How did their experiences differ if at all, did they refer to having had the virus – perhaps you could explore this further in results and discussion.

This is a good point, we have addressed this in the results section, for example by using a quotation by one of the persons who had survived COVID-19 (page 6, paragraph 2). Since we were not exploring how the virus had affected the participants, but rather how the pandemic-related restrictions had influenced their everyday lives, we did not specifically ask about how the infection was experienced.

p.5 line 49 (Participants and data collection) – Please could you give more detail on how the sample was obtained – It seems that this was a convenience rather than a random sample and it is not clear whether all eligible participants were approached and how many of them said yes.

This is a very valid point, and upon reflection, we have realised that the sample was perhaps not so random as we intended it to be. We have tried to clarify the recruitment of the participants on page 4, paragraph 4.

Did the participants all have capacity to give informed consent – How did you ascertain whether residents had capacity please could you give more detail here.

We have added information that the staff made a professional assessment of each person’s cognitive ability. In addition, we have added information on the interviewers’ assessment before commencement of the interview. These revisions can be found on page 4, paragraph 4.

Some of the results presented (such as the ages and description of the background and demographics of the people interviewed would perhaps work better as part of the results (separated from the methods as you are describing the participants rather than what you did) and it would work well at the start of your results to situate your sample.

Thank you for this comment. However, we have chosen to follow the checklist “Standards for reporting qualitative research, S12, which is why the participants’ characteristics are still described in the methods section on page 4, paragraph 5.

Also, as you are interested in frailty and the ageing body, could you describe the sample a little more, did they have dementia or other long term conditions, again this provides useful context for your results.

This is a very relevant comment, but unfortunately, we did not have access to information on the objective health status of the participants. To try to describe the sample, we have added brief information on the health status of people living in Swedish nursing homes on a general level in the introduction section, page 3, paragraph 1.

p.6 Data analysis section – How did you resolve discrepancies between the different researchers conducting the thematic analysis if there were any?

We have extended the description of how the data analysis was conducted, trying to clarify each authors’ role in the analysis procedure and how discrepancies were resolved. This can be found on page 5, paragraph 2.

Results:

Overall in reading the results I was disappointed that there was a lack of detail and depth in relation to the data you present. For each of the subthemes you make interesting assertions however these do

not seem to be backed up by quotations from the participants and where there are quotations they do not necessarily seem to elucidate the richness that you allude to in your narrative account. This is the case for all of the subthemes and was my main frustration when reading the manuscript.

I was struck that you make comments like “this made it possible to defy both the pandemic and the ageing body” but it is not supported by evidence. You refer to frailty and the body at various points but I think need to give a richer, more in depth analysis or if the data is not available to support your assertions then it should be taken out.

We are very grateful for these comments, they have really helped us to improve the manuscript. We have gone through all data again, to deepen our interpretation and revise the sub-themes to make more justice to the richness of the data. We have also exchanged two of the quotations to try to back up our findings in a clearer way. The revised results can be found on page 5-7.

I was interested in how the subthemes “feeling taken care of” and “being in the hands of others” related to each other – Again I wanted to hear more about these different positions, presumably the participants were referring to staff here taking this dual role of both supporter and enforcer. Perhaps you could say something more about this.

This too is a brilliant comment that has helped us deepen the interpretation of our data. The sub-theme “Being in the hands of others” has now been revised to “Having limited freedom” and we have tried to clarify how all the sub-themes relate to each other and to the overarching theme (page 5-7)

Discussion:

Your discussion is interesting and you give a good summary of the strengths and weaknesses.

p. 8 line 30 “Another question mark is whether the pandemic presents an actual risk to the health of older persons in nursing homes” I may have misunderstood but I am not sure what is meant by this comment – perhaps it could be reworded or explained differently as I think in the introduction you present evidence of the disproportionate impact of the pandemic upon nursing home residents both in Sweden and Globally.

Thank you for making us aware of the unclarity in our expression. We have now revised this section to clarify what we meant. The revision is found in the last paragraph on page 7.

I think some of your assertions and interpretation is again limited by the lack of richness in your thematic analysis. For example how do your findings relate to the assertion you make about Dichter et al study – these opposite ends of the spectrum seem to relate to two of your quotes but how do they relate to each other.

We have tried to clarify this in a two-step procedure; first, we revised the results through a deeper analysis (page 5-7). Second, we revised the discussion on person-centred care, trying to clarify the importance of both protecting people from the virus and attending to personal desires and needs. The revision of our assertion about Dichter et al's study can be found on page 8, paragraph 2. We have also made some additional revisions to the discussion, to clarify and discuss our deepened interpretation of the data.

p.9 line 30 “Contrary to expectations, the findings illustrate how the older persons maintained many daily routines, without fear for the virus, and that they avoided pandemic worry by living one day at a time.” – This may well be because there were tight restrictions in place and that they were able to live without fear because of the restrictions in place. I agree with your comment however I think that perhaps you need to relate this to the particular context of your study.

We have revised the sub-theme “living one day at a time, without fear of the virus” (page 6, paragraph 1 and 2), and the sub-theme “Feeling taken care of” (page 6, paragraph 3 and 4) to try to clarify how this part of the discussion relates to our study.

Reviewer: 2 [Editor's Note: although this reviewer is experienced in COVID and the elderly it would appear they are not too familiar with qualitative research so some of their suggestions are not

applicable]

Dr. Ignacio Novo-Veleiro, Complejo Hospitalario Universitario de Santiago de Compostela

Comments to the Author:

The authors present a manuscript of some interest but it is not really a research paper but really a summary of 10 interviews to people living in a nursing home.

Thank you for taking your time to read our paper. However, we do not agree that it is merely a summary of 10 interviews since we have followed methodology and practice for qualitative studies. This is verified by our reporting checklist; Standards for reporting qualitative research, and we have added information on qualitative analysis in the methods section, page 5, paragraph 2, and we have added two methodological references.

It only includes 10 patients, which can not represent the main opinion of people living in the selected nursing home. Even as an interview report, this number of participants is too low to consider it as a valuable paper. The authors should include more patients, at least a half of the whole people living in the center to establish any valid conclusion.

As stated in our previous response, we do not agree with this comment. With all due respect, qualitative studies are not about quantity, which we describe in the discussion section of our paper, under Strengths and weaknesses of the study, page 8, paragraph 1.

The manuscript has a structure of an original research item but the results section is only a summary of the opinions of the interviewed people, it does not include any analysis or data that can lead the reader to any conclusion about their research. In my opinion the manuscript can not be published in the present form and could be rewritten as a letter to the editor or brief communication, since it does not fit as a research paper.

Please see previous responses.

The conclusions are unclear and anyway very difficult to sustain with data from only 10 people. Thank you for pointing out that our conclusions were unclear. We have now revised them, please see page 2.

VERSION 2 – REVIEW

REVIEWER	Rapaport, Penny University College London, Division of Psychiatry
REVIEW RETURNED	30-Apr-2021
GENERAL COMMENTS	I am satisfied that the authors have addressed all of my comments on the previous iteration and the paper is much clearer and I believe should be published as an important addition to the literature on Covid 19 and care homes.